The final spawning ground of Tachypleus gigas (Müller, 1785) on the east Peninsular Malaysia is at risk: a call for action

Nelson Bryan Raveen brav_11@hotmail.com 1 2
Satyanarayana Behara satyam2149@gmail.com 3 4 5
Moh Julia Hwei Zhong 1
Ikhwanuddin Mhd 1
Chatterji Anil 6
Shaharom Faizah 7
1 Institute of Tropical Aquaculture (AKUATROP), Universiti Malaysia Terengganu—UMT , Kuala Terengganu , Terengganu , Malaysia
2 Horseshoe Crab Research Group (HCRG), Universiti Malaysia Terengganu—UMT , Kuala Terengganu , Terengganu , Malaysia
3 Mangrove Research Unit (MARU), Universiti Malaysia Terengganu—UMT, Institute of Oceanography and Environment (INOS) , Kuala Terengganu , Malaysia
4 Laboratory of Systems Ecology and Resource Management, Université Libre de Bruxelles—ULB , Brussels , Belgium
5 Laboratory of Plant Biology and Nature Management, Vrije Universiteit Brussel—VUB , Brussels , Belgium
6 Biological Oceanography Division, National Institute of Oceanography (NIO) , Goa , India
7 Institute of Kenyir Research (IPK), Universiti Malaysia Terengganu—UMT , Kuala Terengganu , Terengganu , Malaysia
Esteban María Ángeles
Electronic publication date: 2016 Jul 19
Publication date: 2016
Volume: 4
Electronic Location ID: e2232
Received 2016 Mar 20; Accepted 2016 Jun 17
Copyright: ©2016 Nelson et al.
Copyright year: 2016
Copyright holder: Nelson et al.
License: This is an open access article distributed under the terms of the Creative Commons Attribution License, which permits unrestricted use, distribution, reproduction and adaptation in any medium and for any purpose provided that it is properly attributed. For attribution, the original author(s), title, publication source (PeerJ) and either DOI or URL of the article must be cited.
License URL: https://creativecommons.org/licenses/by/4.0/

Keywords: Anthropogenic disturbance, Conservation and management, Living fossil, Monsoonal impact, Nest shifting behaviour, Seasonal nesting

Funding: Ministry of Higher Education, Malaysia Funding was provided by the Ministry of Higher Education, Malaysia [scholarship (MyBrain-15) to Mr. BRN]. The funders had no role in study design, data collection and analysis, decision to publish, or preparation of the manuscript.

==============================
Tanjung Selongor and Pantai Balok (State Pahang) are the only two places known for spawning activity of the Malaysian horseshoe crab - Tachypleus gigas (Müller, 1785) on the east coast of Peninsular Malaysia. While the former beach has been disturbed by several anthropogenic activities that ultimately brought an end to the spawning activity of T. gigas, the status of the latter remains uncertain. In the present study, the spawning behavior of T. gigas at Pantai Balok (Sites I-III) was observed over a period of thirty six months, in three phases, between 2009 and 2013. Every year, the crab’s nesting activity was found to be high during Southwest monsoon (May–September) followed by Northeast (November–March) and Inter monsoon (April and October) periods. In the meantime, the number of female T. gigas in 2009–2010 (Phase-1) was higher (38 crabs) than in 2010–2011 (Phase-2: 7 crabs) and 2012–2013 (Phase-3: 9 crabs) for which both increased overexploitation (for edible and fishmeal preparations) as well as anthropogenic disturbances in the vicinity (sand mining since 2009, land reclamation for wave breaker/parking lot constructions in 2011 and fishing jetty construction in 2013) are responsible. In this context, the physical infrastructure developments have altered the sediment close to nesting sites to be dominated by fine sand (2.5Xφ ) with moderately-well sorted (0.6–0.7σφ), very-coarse skewed (−2.4SKφ), and extremely leptokurtic (12.6Kφ) properties. Also, increased concentrations of Cadmium (from 4.2 to 13.6 mg kg−1) and Selenium (from 11.5 to 23.3 mg kg−1) in the sediment, and Sulphide (from 21 to 28 µg l−1) in the water were observed. In relation to the monsoonal changes affecting sheltered beach topography and sediment flux, the spawning crabs have shown a seasonal nest shifting behaviour in-between Sites I-III during 2009–2011. However, in 2012–2013, the crabs were mostly restricted to the areas (i.e., Sites I and II) with high oxygen (5.5–8.0 mg l−1) and moisture depth (6.2–10.2 cm). In view of the sustained anthropogenic pressure on the coastal habitats on one hand and decreasing horseshoe crabs population on the other, it is crucial to implement both conservation and management measures for T. gigas at Pantai Balok. Failing that may lead to the loss of this final spawning ground on the east coast of P. Malaysia.

Introduction

The surviving horseshoe crab species, three Asian—Tachypleus gigas (Müller, 1785), Tachypleus tridentatus (Leach, 1819), Carcinoscorpius rotundicauda (Latreille, 1802), and one American—Limulus polyphemus (Linnaeus, 1758) persist from the Ordovician period (Rudkin & Young, 2009). Its body with armour-like carapace, degenerated spines (adult), appendages with setae, spine-like telson, etc., shows their prehistoric appearance clearly (Walls, Berskson & Smith, 2002; Ruppert, Fox & Barnes, 2004; Smith, Millard & Carmichael, 2009). Of the four species, only C. rotundicauda breeds in the muddy areas near mangroves while the rest spawn along the intertidal beaches of the estuarine coasts (Cartwright-Taylor et al., 2011; Nelson et al., 2015). The selective nesting behaviour of the crabs is usually facilitated by appendage setae (chemoreceptors) which can sense and detect suitable sandy/muddy substratum for their egg incubation and hatching (Botton, 2009).

Horseshoe crabs are encountered only when they come ashore to shallow and surf-protected beaches for nesting. A male crab attached to the rear end of a female crab (using their pedipalps) is distinguished as one mating pair or amplex (Brockmann, 2003; Duffy et al., 2006; Brockmann & Smith, 2009). In addition, satellite males - single and available close to the amplexed pairs to fertilize their eggs and, solitary females - if they are alone, are also visible on the beaches (Mattei et al., 2010; Nelson et al., 2015). A female crab is capable of laying 200–300 eggs (Chatterji & Abidi, 1993) in varying depths (10–20 cm) below the sand (Botton, Tankersley & Lovel, 2010). However, its spawning activity is largely governed by season and local environmental (sediment and water) conditions (Smith, 2007; Weber & Carter, 2009). Despite the increased scientific attention on the horseshoe crabs globally (Smith, Millard & Carmichael, 2009; Chatterji & Shaharom, 2009; Mattei et al., 2010; Cartwright-Taylor et al., 2011; Srijaya et al., 2014), their population is consistently decreasing over the years due to natural (e.g., coastal erosion) as well as anthropogenic disturbances (e.g., pollution, overexploitation, etc.) (Jackson, Nordstrom & Smith, 2005; Ngy et al., 2007; Faurby et al., 2010; Nelson et al., 2015). In addition, the delayed maturity (between 9 and 11 years) and longer embryogenesis period (up to 42 days) (Coursey et al., 2003; Chabot & Watson, 2010) are making these crabs vulnerable to recent changes in the coastal environments (Botton et al., 2006; Chatterji & Shaharom, 2009; Nelson et al., 2015). In fact, L. polyphemus is recognised as ‘near threatened’ species, while the remaining three Asian horseshoe crabs are under “data deficient” category in the IUCN red list (IUCN, 2016).

In Malaysia, all three Asian horseshoe crabs are available. While T. gigas and C. rotundicauda are present in the coastal areas of Peninsular Malaysia, the distribution of T. tridentatus is restricted to East Malaysia (i.e., Sabah and Sarawak) (Chatterji et al., 2008). Although several researchers in P. Malaysia have worked on T. gigas, most of their findings are based on short-term investigations (e.g., Zaleha et al., 2010; John et al., 2011; Kamaruzzaman et al., 2011; Tan, Christianus & Satar, 2011; John, Jalal & Kamaruzzaman, 2013). The only long-term (2009–2011) study that examined the nesting behaviour of T. gigas was carried out by Nelson et al. (2015) from Tanjung Selongor. In fact, Tanjung Selongor and Pantai Balok (in State Pahang) are the only two places known for T. gigas spawning on the east coast of P. Malaysia. According to Nelson et al. (2015), the physical infrastructural developments such as jetty and road/bridge constructions at Tanjung Selongor have already brought an end to the spawning activity of T. gigas. In the case of Pantai Balok, Zaleha et al. (2012), Tan et al. (2012), and John et al. (2012) have observed the spawning populations and nesting behaviour of T. gigas, but for different months in 2009–2010 with no seasonal cross-checking. Therefore, an assessment on the seasonal impact as well as state-of-the-art information on T. gigas is necessary and still to be ascertained from Pantai Balok.

The present study was aimed at investigating the relationship between the nesting activity of T. gigas and the environmental (water/sediment) conditions noticed at Pantai Balok. In specific, identification of major environmental factors predisposed by lunar and monsoonal changes that support T. gigas spawning formed the main forte of this study. In recent years, several anthropogenic activities such as sand mining (2009 - to present), land reclamation (2011), and construction of a fishing jetty (2013) also appeared to influence T. gigas population. At the end, a few recommendations were offered for possible conservation and management of T. gigas at Pantai Balok.

Materials and Methods

Study area

We recall that Pantai Balok, in State Pahang, is one of the two spawning grounds for T. gigas on the east coast of P. Malaysia (Lat: 3°56′16.58″-3°55′39.33″N; Long: 103°22′32.74″-103°22′27.12″E) (Fig. 1A). River Balok and its tributaries opens here into South China Sea and provide a regular exchange of water (Fig. 1B). The climate of Pantai Balok is influenced by Northeast (NE) (November–March) and Southwest (SW) monsoons (May–September), separated by two Inter-monsoon (IM) periods (April and October). The weather—with a temperature varying between 20 and 36 °C, is generally hot and humid (WU, 2014). The annual (average) rainfall is about 1710.5 mm and occurs mostly during September-December. The tides with a range of 0.1–3.4 m are mixed in nature (NHC, 2013).

Figure 1 Study area.

(A) Geographic location of State Pahang and Pantai Balok (square box) on the east coast of Peninsular Malaysia; (B) The sampling sites (I–III) in River Balok estuary along with the locations of sand mining (▴), wave breaker/parking lot construction and, fishing jetty construction (*) in the vicinity (Google Map © 2013). Photographs shows the physical infrastructural developments at Site-I - (C) beach condition in 2009–2010, (D) wave breaker/parking lot construction in 2011 and, (E) fishing jetty construction in 2013.

Pantai Balok is under sustained human intervention over the last few decades. In addition to the sand mining since 2009 (Fig. 1B), both land reclamation for wave breaker and parking lot constructions in 2011 (Fig. 1D), and fishing jetty construction in 2013 (Fig. 1E) have brought considerable changes to the beach topography, sediment and water characteristics (present study). Local people also catch the mating pairs of T. gigas for their food and processed feed preparations (to use in chick and fish farms). In order to carry out the present study, the local fishermen’s association at Pantai Balok was consulted and their permission was obtained.

Sampling protocol

The spawning activity of T. gigas was observed over a period of thirty six months in three phases between 2009 and 2013. Phase-1 corresponds to the observations made from July 2009 to April 2010 (for both full moon and new moon periods), while Phase-2 from June 2010 to June 2011 (only for full moon due to financial limitations), and Phase-3 from May 2012 to May 2013 (for full moon and new moon periods). Overall, the dataset represents 15 months of observations each for SW and NE monsoons and 6 months of observations for IM periods. Restricted to a gentle slope in-between high and mid tide markings, the spawning activity of T. gigas was found only in a portion of 381 m along the beach at Pantai Balok. The places that have shown regular yield of nests were divided into three sampling sites namely, Site-I (3°56′15.76″N, 103°22′33.96″E), Site-II (3°56′12.90″N, 103°22′36.62″E) and, Site-III (3°56′16.30″N, 103°22′40.36″E) (located 105–142 m apart). Although biological and environmental parameters (as outlined below) were obtained from each site for every month, the months that have shown eggs and/or spawning crabs were only considered for the present analysis/interpretation (for complete details, see the Supplemental Information). In order to have a better understanding on the nesting activity of T. gigas vis-à-vis environmental conditions, the pollution indicating factors such as heavy metal (i.e., Lead (Pb), Chromium (Cr), Selenium (Se), Cadmium (Cd), Copper (Cu) and Zinc (Zn)) concentrations in the sediment; nutrient (i.e., Nitrite (NO2−), Nitrate (NO3−) and Phosphate (PO43−)) and Sulphide (S2−) concentrations in the water were also tested (for Phases 2 and 3).

Biological observations

All sampling sites were visited at night/early morning during high tide for spawning crabs estimation and to mark their nesting locations, whereas in daytime during low tide for nest/eggs counting. The spawning (male/female) crabs of T. gigas were counted by sight if they are available ashore and by catch (using hand) if they are submerged in water and releasing the air bubbles. The places that shown air bubbles were marked with wooden stakes for nest/eggs counting. The number of nests was obtained through removing the sand at the point of crab imprints (using plastic hand shovel). The egg clutches from each nest were gently removed and washed in situ with seawater (using 2 mm sieve) to count the number of eggs. After counting, all eggs were placed back into the same pit and covered by sand. However, the data on overexploitation of the crabs by local fishermen were qualitative (learnt from other villagers more than from our observations) and hence used only for the analysis.

Hydrological observations

The data on surface water quality parameters such as temperature (°C), salinity (‰), pH and dissolved oxygen (DO) (mg l−1) were obtained in situ using YSI 556 multi-probe sensors (YSI Inc., Yellow-Springs, OH, USA). In addition, 1,000 ml of water was collected from each site for laboratory analyses. All samples were first filtered through microglass filter paper (Whatman® GF/C 47 mm, England) and then kept in the ice-box. While the filter paper used for each sample was preserved separately in 90% Acetone for Chl-a estimation (Parsons, Maita & Lalli, 1984), the pH of the filtered water was adjusted to 2 by adding hydrochloric acid (HCl) (to arrest microbial activity and stabilize organic carbon) (USEPA, 2004; Wilde et al., 2009). The analyses of nutrients and sulphide were carried out in the laboratory using spectrophotometer kit (HACH DREL 2400, USA) (HACH, 2004; Magarde et al., 2011).

Sedimentological observations

Both soil temperature (using thermometer, sensitivity: ±0.2 °C) and pH (DM-13 Takamura Electric Works, Japan) were recorded in situ at the time of nest/egg counting. A transparent PVC tube with 3″ diameter and 1 m length (marked up to 50 cm) was pushed into the sediment and, the depth above the anoxic/black sand layer was measured as moisture depth at each sampling site. In addition, ∼500 g of surface sediment was collected from each site (using hand shovel) for the laboratory analyses. About 100 g of the sediment was oven dried for 3 days (at 45 °C) and then separated into different fractions (using <63 to 4,000 µm sieves) using a mechanical sieve shaker (Retsch AS 200 Basic, Germany). The logarithmic method of moments was used to estimate mean (φ), sorting (σφ), skewness (SKφ) and kurtosis (Kφ) values in the samples (Blott & Pye, 2001). The sediment fraction <63 µm was used for heavy metal analysis through Inductively Coupled Plasma Optical Emission Spectrometry (ICP-OES) (Varian Vista-Pro, USA) (Noriki et al., 1980). The total organic carbon (TOC) in the samples was estimated using a TOC analyser (Shimadzu, TOC-VCPH, Japan) (Schumacher, 2002; USEPA, 2004). In addition, the sediment enrichment factor (Leong, Kamaruzzaman & Zaleha, 2003) and the geo-accumulation index (Din, 1992) were derived for understanding the impact of pollution in the vicinity.

Statistical analyses

The statistical variations within biological and environmental parameters (at P < 0.05) were tested through One-Way ANOVA (using OriginPro v.9.1 software), for which the data obtained for each (select) parameter from every month in Phases 1-3 were considered. In this context, the phase-wise information was treated as (1-3) dependent groups and tested against to the samplings (I-III) sites, monsoon (SW, NE and IM) seasons, and lunar (full moon and new moon) periods as independent groups. The Principle Component Analysis (PCA) was also used to identify the percentage (%) variation between environmental (sediment and water) parameters and T. gigas egg counts (root-transformed data) (using PRIMER v.6 software) (Clarke & Gorley, 2006). To establish the species-environment relationship, a routine called BEST - amalgamating BIO-ENV and BVSTEP procedures, in PRIMER v.6 was followed (with 999 permutations). In this context, the Spearman’s rank correlation coefficients (ρ values) were considered for denoting positive and negative impact of the environmental variables on T. gigas nesting sites.

Results

Spawning population and nesting

Out of thirty-six months investigation, the spawning activity of T. gigas was found only for twenty-two months, especially during SW monsoon followed by NE and IM periods (Tables 1–5) (Fig. 2A). The full moon observations revealed higher egg/nest yield (102–208 nos.) than the new moon observations (82–121 nos.) (Fig. 2B), where the differences were non-significant (Table 6). In total, 117 spawning crabs (males: 63 and females: 54) were recorded for the entire period of investigation. Despite the highest number of spawning crabs at Site-II, Site-III showed maximum egg/nest yield followed by Site-I (Fig. 2C). Although number of the spawning crabs was higher for 2009–2010 (Phase-1: 69 crabs), it decreased during 2010–2011 (Phase-2: 20 crabs) and 2012–2013 (Phase-3: 28 crabs) (Fig. 2D). Another important observation is that the female crabs dug more number of nests (up to 32) and released more number of eggs (3.977 nos. in 43 clutches) in Phase-3 than to Phase-1 (23 nests and 3,025 eggs in 28 clutches) or Phase-2 (9 nests, 1,952 eggs in 10 clutches) (Tables 1–5). Also, the number of amplexed pairs decreased from 12 in Phase-1 to 7 in Phase-2 and 4 in Phase-3, along with the solitary females from 25 in Phase-1 to 5 in Phase-3. At the time of less anthropogenic intervention (i.e., Phase-1), T. gigas displayed a seasonal nest shifting behaviour from open coastal area (Sites II-III) in SW monsoon to sheltered beach (Site-I) in NE and IM periods (Tables 1–2) (Fig. 6).

Table 1 Ecobiological observations from the three (I-III) nesting sites of T achypleus gigas at Pantai Balok during Phase-1 (2009–2010) full moon surveys.

	2009	2010	
	July	August	October	November	March	April	
	I	II	III	I	II	III	I	II	III	I	II	III	I	II	III	I	II	III	
(A) Biology	
Nest (nos.)	–	–	3	–	3	–	2	–	–	1	–	–	2	–	–	4	–	–	
Egg (nos.)	–	–	502	–	318	–	340	–	–	114	–	–	237	–	–	455	–	–	
Clutches (nos.)	–	–	4	–	3	–	2	–	–	1	–	–	3	–	–	5	–	–	
Male (nos.)	1	4	–	–	1	2	1	2	–	2	–	–	1	1	–	1	2	–	
Female (nos.)	–	2	4	2	2	4	4	–	–	2	1	–	2	–	–	–	–	3	
(B) Sediment	
Mean (Xφ)	2.1	1.4	1.1	1.8	1.1	1.5	1.1	2.4	1.7	1.2	2.5	2.0	1.2	2.1	2.4	1.0	2.1	2.1	
Sorting (σφ)	1.0	0.9	1.1	1.0	1.0	1.2	0.9	0.5	1.4	0.8	0.7	1.7	1.0	0.9	0.9	1.0	1.0	1.1	
Skewness (SKφ)	0.5	0.5	0.1	0.2	0.0	0.0	0.0	0.1	0.3	0.0	0.2	0.6	0.0	0.1	0.7	0.0	0.1	0.4	
Kurtosis (Kφ)	3.3	2.5	2.5	3.0	3.9	2.3	2.5	4.6	2.4	2.1	4.4	2.2	2.6	3.9	2.7	2.7	3.7	3.1	
Gravel (%)	1.3	0.8	2.4	2.6	2.8	1.8	2.8	0.0	0.5	2.2	0.1	0.4	2.4	0.4	1.9	2.1	0.4	2.3	
Sand (%)	93.6	97.1	94.7	92.8	94.6	95.1	93.7	99.5	95.2	94.6	99.7	94.5	94.6	98.4	94.0	94.7	97.5	94.1	
Silt & Clay (%)	5.1	2.1	2.9	4.7	2.6	3.1	3.5	0.5	4.2	3.3	0.1	5.1	3.0	1.1	4.2	3.2	2.2	3.6	
0.125 mm (%)	4.7	2.2	1.7	3.2	1.4	2.4	1.9	6.0	4.1	1.9	5.3	4.3	1.9	3.2	3.2	1.7	3.6	2.7	
0.180 mm (%)	16.1	9.8	11.4	14.6	10.6	11.3	12.3	15.6	15.6	11.6	16.0	16.4	12.4	14.9	14.6	11.4	13.4	13.7	
0.250 mm (%)	22.3	20.2	19.5	21.7	19.0	20.1	19.5	20.7	25.8	19.6	22.5	27.2	19.7	24.5	22.7	19.5	22.7	20.3	
Moisture depth (cm)	3.2	5.7	4.8	4.1	4.6	4.3	3.5	2.8	2.4	2.7	1.9	2.2	5.1	5.6	5.9	6.7	6.3	6.8	
Temperature (°C)	32.2	31.2	32.7	30.8	31.2	31.5	30.1	29.9	29.9	28.7	29.1	28.8	30.2	30.8	30.6	32.5	32.0	32.5	
pH	6.1	6.1	5.9	6.2	6.5	6.5	6.5	6.6	6.6	6.3	6.1	6.4	5.8	5.9	5.8	5.8	5.2	5.5	
(C) Water	
Temperature (°C)	29.4	29.7	29.5	30.0	29.6	29.5	32.9	33.2	33.4	29.3	29.4	29.5	29.5	29.7	30.1	31.2	30.8	30.5	
pH	7.8	7.5	7.5	7.7	7.7	7.8	7.8	7.1	7.6	8.2	7.6	7.7	6.9	7.1	6.7	7.2	7.3	7.0	
Salinity (‰)	28.7	31.0	32.1	32.5	33.5	32.0	34.2	38.4	31.8	31.7	26.4	28.8	37.2	39.2	40.4	34.6	37.8	38.5	
DO (mg l−1)	5.4	5.1	5.0	4.3	5.4	6.0	5.0	3.5	5.5	6.2	4.9	6.0	5.8	5.2	6.1	5.6	5.7	6.0	
Notes.

“–” No sample observed at the time of investigation.

Figure 2 Egg/nest yield of Tachypleus gigas at Pantai Balok in relation to - (A) season, (B) lunar period, (C) sampling sites and, (D) the number of male and female spawning crabs arriving at Balok beach.

Table 2 Ecobiological observations from the three (I–III) nesting sites of T achypleus gigas at Pantai Balok during Phase-1 (2009–2010) new moon surveys.

	2009	2010	
	August	October	November	March	April	
	I	II	III	I	II	III	I	II	III	I	II	III	I	II	III	
(A) Biology	
Nest (nos.)	–	1	–	2	–	–	1	–	–	2	–	–	2	–	–	
Egg (nos.)	–	104	–	385	–	–	92	–	–	158	–	–	320	–	–	
Clutches (nos.)	–	1	–	3	–	–	1	–	–	2	–	–	3	–	–	
Male (nos.)	–	2	1	–	–	–	1	–	–	1	4	1	–	2	1	
Female (nos.)	2	–	–	–	–	–	2	–	–	–	1	4	3	–	–	
(B) Sediment	
Mean (Xφ)	1.9	1.3	1.3	1.2	2.1	1.7	1.2	2.4	1.9	1.1	2.1	2.2	1.1	1.9	1.9	
Sorting (σφ)	1.0	1.0	1.2	1.0	0.7	1.4	0.9	0.6	1.5	1.1	0.8	1.1	1.1	1.0	1.0	
Skewness (SKφ)	0.3	0.1	0.0	0.1	0.3	0.3	0.0	0.2	0.5	0.0	0.1	0.5	0.0	0.1	0.5	
Kurtosis (Kφ)	3.1	3.3	2.3	2.7	4.0	2.3	2.4	4.5	2.3	2.7	3.7	2.9	3.1	3.4	3.3	
Gravel (%)	2.0	2.4	2.2	2.8	0.4	0.8	2.8	0.1	0.4	2.3	0.4	2.0	1.9	0.5	2.8	
Sand (%)	93.2	95.9	94.8	93.5	98.4	95.0	93.9	99.6	94.8	94.9	97.9	94.1	94.6	96.9	93.1	
Silt & Clay (%)	4.8	1.8	3.0	3.7	1.2	4.2	3.3	0.3	4.7	2.9	1.7	3.9	3.5	2.6	4.2	
0.125 mm (%)	4.1	1.8	2.1	2.3	5.4	3.7	1.8	5.8	4.2	1.6	3.3	2.8	2.7	4.2	3.0	
0.180 mm (%)	15.3	10.0	11.4	13.7	14.6	14.7	11.9	15.8	16.0	11.8	14.0	14.1	11.9	14.7	13.0	
0.250 mm (%)	21.9	19.4	19.7	20.0	20.2	23.9	19.7	21.5	26.3	19.6	23.2	21.6	18.5	20.4	18.7	
Moisture depth (cm)	3.7	5.1	4.5	3.7	3.0	3.1	3.1	2.3	2.3	5.9	5.9	6.3	6.2	6.5	6.5	
Temperature (°C)	32.6	32.4	32.5	30.0	30.2	30.5	29.3	29.3	29.5	29.9	30.0	30.2	32.3	32.3	32.5	
pH	6.2	6.2	6.1	6.2	6.3	6.4	6.5	6.7	6.7	6.0	5.8	5.9	5.5	5.4	5.3	
(C) Water	
Temperature (°C)	29.7	29.6	29.5	31.1	31.3	31.3	31.0	31.2	31.4	30.3	30.2	30.3	31.6	31.3	31.0	
pH	7.7	7.6	7.7	7.6	7.2	7.6	8.0	7.4	7.7	7.0	7.2	6.8	7.1	7.5	7.3	
Salinity (‰)	30.6	32.3	32.1	35.4	37.1	31.9	32.9	32.1	30.2	35.9	38.6	39.4	36.6	35.4	36.7	
DO (mg l−1)	4.8	5.2	5.5	4.6	4.2	5.0	5.6	4.2	5.8	5.7	5.5	6.0	5.9	6.0	5.7	
Notes.

“–” No sample observed at the time of investigation.

Sediment characteristics

The beach sediment that was largely represented by medium sand (1–2Xφ) in Phase-1 (Tables 1–2) was changed into fine sand (>2Xφ ) in Phases 2 and 3 (Tables 3–5). Categorically, the sediment fraction that contained 0.250 mm (representing medium sand) was more in Phase-1, whereas it replaced by 0.180 mm (representing medium-fine sand) in Phase-2 and 0.125 mm (representing fine sand) in Phase-3. For Phase-1, the sediment was represented by moderately-well sorted to poorly sorted (0.5 − 1.7σφ), symmetrical to very-fine skewed (0.0 − 0.7SKφ), and very leptokurtic to extremely leptokurtic (2.1 − 4.6Kφ) properties (Tables 1–2). In the case of Phase-2, except skewness (i.e., fine skewed to very-fine skewed: 0.1 − 3.1SKφ), both sorting (0.5 − 1.4σφ) and kurtosis (2.5 − 20.5Kφ) remained as same as Phase-1 (Table 3). The well-sorted to moderately sorted (0.4 − 1.0σφ), very-coarse skewed (−3.1 − − 1.1SKφ), and extremely leptokurtic (7.8 − 16.8Kφ) properties have characterised the sediment collected for Phase-3 (Tables 4–5). Although there was not much variation in the moisture depth between Phase-1 and Phase-2 (average, 4.2–4.4 cm), it was rather increased in Phase-3 (to 6.8 cm). A significant decrease in (average) silt and clay, and pH measurements was observed between Phase-1 and Phase-3 (Table 6). The changes in TOC between Phase-2 and Phase-3 were insignificant, except for SW vs. NE monsoon (Table 6).

Table 3 Ecobiological observations from the three (I–III) nesting sites of T achypleus gigas at Pantai Balok during Phase-2 (2010–2011) full moon surveys.

	2010	2011	
	June	July	August	October	March	April	May	June	
	I	II	III	I	II	III	I	II	III	I	II	III	I	II	III	I	II	III	I	II	III	I	II	III	
(A) Biology	
Nest (nos.)	–	–	–	–	–	2	–	2	–	–	–	–	3	–	–	1	–	1	–	–	–	–	–	–	
Egg (nos.)	–	–	–	–	–	581	–	455	–	–	–	–	556	–	–	342	–	18	–	–	–	–	–	–	
Clutches (nos.)	–	–	–	–	–	3	–	2	–	–	–	–	3	–	–	1	–	1	–	–	–	–	–	–	
Male (nos.)	–	1	1	–	3	–	–	2	–	–	1	–	2	–	–	1	–	–	–	–	1	–	–	1	
Female (nos.)	–	1	1	–	2	–	–	1	–	–	–	–	–	–	–	–	–	–	–	–	1	–	–	1	
(B) Sediment	
Mean (Xφ)	1.9	1.8	1.9	1.7	1.7	2.1	1.2	1.6	0.6	0.0	0.0	0.0	2.0	2.0	1.6	2.4	2.6	2.3	2.2	2.2	2.3	2.2	2.2	2.2	
Sorting (σφ)	1.1	1.1	1.1	1.0	1.0	0.9	1.4	1.1	1.3	0.5	0.5	0.6	0.9	1.1	1.2	0.5	0.5	0.5	0.6	0.6	0.6	0.7	0.7	0.8	
Skewness (SKφ)	1.7	1.6	1.6	1.1	0.7	1.8	0.1	0.7	0.5	1.4	0.9	2.1	1.9	1.4	0.8	2.4	2.5	2.2	2.6	2.6	2.7	3.1	2.9	2.5	
Kurtosis (Kφ)	5.5	5.1	5.3	4.5	3.9	6.8	3.5	3.6	2.5	6.5	4.9	8.8	7.3	5.3	2.7	19.8	20.5	17.4	14.6	14.0	14.0	15.4	13.3	10.7	
Gravel (%)	5.0	5.2	4.3	3.1	1.9	1.4	7.7	3.2	10.2	0.6	0.7	0.5	2.3	3.0	3.2	0.3	0.3	0.3	1.3	0.9	0.6	1.3	1.7	2.1	
Sand (%)	94.3	94.3	95.0	95.5	96.0	97.2	88.7	93.5	86.8	99.4	99.3	99.5	97.4	96.3	95.7	99.2	99.2	99.4	98.5	98.7	99.0	98.6	98.0	97.5	
Silt & Clay (%)	0.6	0.5	0.7	1.4	2.1	1.4	3.6	3.3	3.0	0.0	0.0	0.0	0.4	0.8	1.1	0.5	0.5	0.3	0.3	0.3	0.4	0.1	0.2	0.3	
0.125 mm (%)	18.8	17.1	18.5	13.9	9.6	11.9	5.9	7.7	2.6	0.1	0.1	0.2	14.3	14.8	13.3	17.6	16.2	11.7	9.3	8.3	7.8	11.4	10.7	9.7	
0.180 mm (%)	28.5	26.1	27.8	28.4	29.6	60.5	25.7	26.3	18.6	0.3	0.2	0.6	47.5	43.7	34.0	66.1	62.9	66.7	63.4	65.7	70.6	71.5	69.3	64.8	
0.250 mm (%)	41.3	40.5	37.7	26.5	15.9	7.4	14.8	16.5	9.3	2.0	0.3	3.9	18.5	15.2	9.9	6.4	10.6	12.8	16.6	13.7	11.3	8.3	10.3	12.0	
Moisture depth (cm)	6.3	5.2	6.7	4.3	4.8	6.1	3.6	4.4	5.7	3.7	4.1	5.2	2.5	3.7	3.2	2.8	4.3	2.7	4.3	3.2	2.5	4.8	3.6	3.1	
Temperature (°C)	34.4	32.3	32.7	32.5	31.8	31.6	32.8	32.4	32.2	31.3	30.7	31.2	36.5	36.1	35.9	36.3	36.3	36.5	35.3	35.7	35.5	33.8	33.6	33.8	
pH	6.6	6.2	6.3	6.1	6.1	6.0	6.5	6.3	6.5	6.4	6.4	6.4	6.5	6.6	6.5	4.1	4.2	4.1	3.5	3.4	3.4	4.2	4.2	4.0	
Total Organic Carbon (%)	0.2	0.2	0.2	0.1	0.1	0.2	0.1	0.1	0.1	0.2	0.2	0.2	0.1	0.1	0.2	0.1	0.1	0.1	0.1	0.2	0.1	0.1	0.2	0.2	
Cd (mg kg−1)	4.6	3.3	5.4	4.4	3.6	4.0	3.9	3.6	3.3	1.9	2.5	1.5	5.5	5.7	5.2	6.6	7.0	6.0	5.5	3.7	2.0	6.2	4.1	1.8	
Cr (mg kg−1)	30.9	27.4	31.0	29.0	27.6	27.0	26.7	24.0	22.7	19.9	21.5	20.6	33.0	31.2	25.3	29.7	29.6	23.8	25.8	23.1	21.3	24.1	25.5	26.0	
Cu (mg kg−1)	6.6	8.7	3.8	6.2	8.8	8.9	17.3	15.0	15.3	7.1	7.2	7.7	2.7	1.9	0.9	2.6	2.8	2.5	3.3	2.1	0.9	3.2	2.2	1.1	
Pb (mg kg−1)	5.2	4.6	5.1	6.3	7.6	9.1	10.4	9.3	8.9	7.6	8.4	7.7	11.0	12.9	13.0	10.3	12.4	12.1	9.5	9.4	9.7	8.7	9.7	10.4	
Se (mg kg−1)	9.8	8.3	10.3	8.1	6.0	9.3	14.4	11.2	13.9	8.3	8.6	8.9	14.0	14.3	12.6	17.0	18.4	16.4	6.2	6.3	6.6	14.8	15.6	15.9	
Zn (mg kg−1)	32.8	29.5	32.6	27.2	22.5	23.9	32.2	27.5	28.7	18.0	18.7	19.3	21.6	20.7	17.0	20.7	22.3	19.7	15.6	14.9	14.8	17.8	18.8	19.2	
(C) Water	
Temperature (°C)	29.4	29.5	29.5	29.5	29.6	29.5	30.0	30.0	29.8	29.8	29.6	29.6	30.4	30.2	30.3	29.8	29.9	29.7	30.3	30.4	30.4	30.0	30.1	30.1	
pH	7.8	7.5	7.2	7.3	7.6	7.5	7.6	7.8	7.4	7.1	7.5	7.1	6.7	6.2	5.9	6.9	6.5	6.3	8.1	7.7	7.5	8.8	8.4	8.2	
Salinity (‰)	32.1	32.8	33.6	32.0	32.8	33.6	32.5	32.9	33.3	6.6	10.7	12.0	8.2	9.4	10.2	4.1	5.6	6.4	11.1	12.6	13.3	14.9	15.3	17.0	
Dissolved Oxygen (mg l−1)	5.4	5.3	5.3	6.0	5.9	5.8	4.3	4.7	4.8	4.0	3.7	3.4	3.1	3.2	3.4	5.3	5.2	5.2	2.8	3.0	3.1	3.3	3.3	3.3	
NO2− (mg l−1)	0.0	0.0	0.0	0.1	0.1	0.1	0.1	0.1	0.1	0.0	0.0	0.0	0.1	0.1	0.1	0.0	0.0	0.0	0.0	0.0	0.0	0.0	0.0	0.0	
NO3− (mg l−1)	0.9	1.3	1.4	1.4	1.6	1.5	1.1	1.1	0.8	0.3	0.6	0.3	1.5	1.6	1.4	1.1	1.9	1.8	0.8	0.5	0.9	0.6	0.2	0.2	
PO43− (mg l−1)	0.3	0.3	0.3	0.3	0.3	0.3	0.4	0.4	0.4	0.3	0.3	0.4	0.4	0.4	0.4	0.1	0.1	0.1	0.4	0.4	0.4	0.4	0.4	0.4	
S2−(µg l−1)	1.3	1.7	1.0	4.0	4.7	6.0	11.7	7.0	12.7	15.0	9.0	12.7	31.3	28.7	29.7	68.0	66.0	63.3	14.3	12.7	14.0	27.3	29.0	26.3	
Chl-a (mg l−1)	0.1	0.1	0.1	0.1	0.1	0.1	0.0	0.0	0.0	0.1	0.1	0.1	0.1	0.1	0.1	0.0	0.0	0.0	0.1	0.1	0.1	0.1	0.1	0.1	
Notes.

“–” No sample observed at the time of investigation.

Table 4 Ecobiological observations from the three (I–III) nesting sites of T achypleus gigas at Pantai Balok during Phase-3 (2012–2013) full moon surveys.

	2012	2013	
	May	June	July	August	February	March	April	
	I	II	III	I	II	III	I	II	III	I	II	III	I	II	III	I	II	III	I	II	III	
(A) Biology	
Nest (nos.)	–	6	–	–	–	–	8	4	–	1	–	–	2	–	–	–	1	1	–	1	–	
Egg (nos.)	–	868	–	–	–	–	1,074	613	–	254	–	–	80	–	–	–	108	314	–	32	–	
Clutches (nos.)	–	5	–	–	–	–	10	6	–	2	–	–	3	–	–	–	1	3	–	1	–	
Male (nos.)	1	2	–	–	2	–	1	3	1	–	–	–	1	–	–	–	–	–	–	1	1	
Female (nos.)	–	1	1	–	–	1	–	–	2	–	–	–	–	–	–	–	–	–	–	1	–	
(B) Sediment	
Mean (Xφ)	2.5	2.5	2.5	2.4	2.5	2.6	2.4	2.4	2.3	2.4	2.4	2.3	2.5	2.4	2.5	2.4	2.4	2.4	2.6	2.6	2.6	
Sorting (σφ)	0.6	0.7	0.6	0.9	0.9	0.8	0.8	1.0	0.9	0.8	0.9	0.9	0.6	0.7	0.5	0.7	0.8	0.6	0.5	0.6	0.4	
Skewness (SKφ)	−2.7	−2.8	−2.8	−2.7	−2.6	−3.1	−2.7	−2.2	−2.2	−2.6	−2.5	−2.6	−2.0	−1.8	−1.1	−2.2	−2.3	−2.2	−2.1	−2.3	−1.8	
Kurtosis (Kφ)	14.9	14.0	15.7	11.3	10.8	14.6	11.7	8.4	7.8	11.2	9.9	10.4	10.7	8.3	8.6	9.9	10.4	12.3	13.3	12.6	13.8	
Gravel (%)	0.4	1.8	0.4	3.4	5.1	2.4	2.7	3.3	2.0	2.2	2.7	2.8	0.4	0.6	0.2	0.7	1.8	0.7	0.4	0.5	0.2	
Sand (%)	99.4	97.6	99.1	96.3	94.1	97.0	96.8	96.0	97.6	97.4	96.5	96.8	99.1	98.3	99.4	98.5	97.3	99.1	99.2	97.9	99.4	
Silt & Clay (%)	0.2	0.6	0.5	0.4	0.8	0.6	0.4	0.7	0.4	0.4	0.8	0.4	0.5	1.2	0.3	0.8	0.9	0.3	0.5	1.7	0.4	
0.125 mm (%)	40.7	37.5	39.4	34.9	41.9	46.6	39.2	34.6	36.0	35.4	37.2	34.0	29.9	28.5	29.5	37.2	35.4	26.7	39.9	37.7	39.3	
0.180 mm (%)	40.8	44.2	41.1	40.0	30.1	30.3	36.1	32.9	37.0	40.3	33.2	41.4	39.8	34.5	50.3	35.7	35.2	49.6	39.8	32.2	44.8	
0.250 mm (%)	7.7	7.5	7.6	10.5	5.1	4.2	7.8	9.3	7.2	8.2	7.9	9.1	11.3	13.0	10.6	8.6	9.3	13.6	9.0	8.5	5.5	
Moisture depth (cm)	6.0	6.1	6.8	4.9	5.0	4.4	7.1	6.4	7.5	6.3	6.4	9.1	8.4	8.4	7.4	6.2	6.2	6.3	10.1	10.2	10.6	
Temperature (°C)	30.0	28.1	28.9	31.7	32.8	32.3	28.6	32.4	33.5	33.8	32.5	31.4	30.5	28.8	29.4	33.5	33.8	33.7	39.1	37.3	36.6	
pH	5.9	5.8	6.1	6.8	6.5	6.8	4.0	5.4	4.6	3.0	3.8	2.8	5.6	5.6	5.8	4.9	4.9	4.9	2.8	3.2	4.0	
Total Organic Carbon (%)	0.1	0.2	0.2	0.2	0.2	0.1	0.2	0.1	0.1	0.2	0.1	0.1	0.1	0.1	0.1	0.1	0.1	0.1	0.1	0.0	0.0	
Cd (mg kg−1)	15.7	13.3	2.7	18.5	16.4	11.9	26.7	16.8	14.7	9.3	11.1	15.5	9.9	15.4	8.0	20.0	16.1	8.1	21.8	9.6	13.3	
Cr (mg kg−1)	28.8	17.0	17.9	37.7	10.7	13.5	22.4	43.7	10.7	11.0	9.6	15.6	13.7	12.2	13.8	33.3	21.6	29.0	16.0	29.3	16.0	
Cu (mg kg−1)	2.9	2.2	2.4	2.2	2.3	1.6	2.1	3.7	2.1	5.9	2.5	4.0	2.5	4.3	2.9	10.1	3.4	7.2	19.0	7.1	14.7	
Pb (mg kg−1)	6.4	7.8	5.1	5.6	7.4	4.9	4.1	4.7	4.8	8.3	8.3	10.8	23.3	12.7	19.1	20.7	12.9	16.2	20.7	13.5	16.8	
Se (mg kg−1)	29.4	32.1	18.2	27.8	26.3	14.2	27.4	28.0	17.0	18.2	19.9	20.9	15.5	15.2	28.0	36.6	23.4	10.1	21.3	28.0	19.7	
Zn (mg kg−1)	19.0	20.8	15.0	14.6	19.2	14.6	12.3	14.0	14.1	20.6	23.5	27.5	23.8	23.2	23.6	41.6	28.9	30.0	26.8	26.3	27.3	
(C) Water	
Temperature (°C)	27.6	27.6	27.5	29.9	29.6	30.1	30.9	31.3	30.7	30.0	30.1	30.6	28.5	28.7	27.7	30.8	31.1	30.9	32.8	31.6	30.6	
pH	6.2	6.5	6.6	8.4	8.4	8.4	8.0	8.1	7.6	7.9	8.4	8.4	7.9	7.7	7.4	8.6	8.7	8.7	8.2	7.9	7.4	
Salinity (‰)	3.1	2.7	2.3	33.3	33.4	33.3	34.5	34.5	34.6	30.0	31.0	31.2	3.7	3.9	5.4	20.9	21.7	21.7	16.9	15.0	16.7	
Dissolved Oxygen (mg l−1)	2.8	3.2	4.1	5.3	5.1	5.6	6.8	5.4	6.7	7.8	5.8	9.1	7.6	5.0	7.9	5.3	6.4	4.5	4.6	6.0	5.5	
NO2− (mg l−1)	0.0	0.0	0.0	0.0	0.0	0.0	0.0	0.0	0.0	0.0	0.0	0.0	0.1	0.1	0.0	0.0	0.1	0.0	0.1	0.1	0.1	
NO3− (mg l−1)	0.2	0.3	0.3	0.7	1.4	0.3	0.7	0.3	0.6	3.4	0.6	0.7	0.4	0.6	0.7	2.9	1.7	2.4	0.6	0.7	0.9	
PO43− (mg l−1)	0.1	0.2	0.2	0.0	0.1	0.1	0.1	0.1	0.1	0.0	0.1	0.1	0.2	0.2	0.1	0.2	0.2	0.1	0.2	0.2	0.2	
S2− (µg l−1)	47.7	42.3	50.3	20.0	16.3	10.3	5.3	7.3	7.7	39.3	97.3	124.7	23.3	22.0	66.7	0.0	4.7	5.0	58.7	35.7	22.0	
Chl-a (mg l−1)	0.2	0.5	0.2	0.4	0.5	0.4	0.4	0.3	0.5	0.7	0.9	0.5	0.6	0.9	0.5	0.4	0.6	0.5	1.0	1.2	1.0	
Notes.

“–” No sample observed at the time of investigation.

In terms of the season, also both fine and medium-fine sand contents (with moderately to moderately-well sorted nature) were high for NE and IM periods whereas more gravel and silt and clay (with poor to moderately sorted nature) for SW monsoon (Tables 1–5). The three sampling sites that represented largely by medium sand (average, 1.4 − 1.8Xφ) and moderate sorting (0.8 − 1.0σφ) characteristics in Phases 1-2 were replaced by fine sand (2.5Xφ) with moderately-well sorted sediment (0.6 − 0.7σφ) in Phase-3. While its skewness decreased from symmetrical (0.1SKφ) to very-coarse skewed (−2.4SKφ), the kurtosis increased from very leptokurtic (2.7Kφ) to extremely leptokurtic (12.6Kφ) between Phase-1 and Phase-3 (Tables 1–5).

Table 5 Ecobiological observations from the three (I–III) nesting sites of T achypleus gigas at Pantai Balok during Phase-3 (2012–2013) new moon surveys.

	2012	2013	
	May	June	July	February	March	May	
	I	II	III	I	II	III	I	II	III	I	II	III	I	II	III	I	II	III	
(A) Biology	
Nest (nos.)	–	–	–	–	2	–	–	–	–	–	2	–	4	–	–	–	–	–	
Egg (nos.)	–	–	–	–	169	–	–	–	–	–	47	–	418	–	–	–	–	–	
Clutches (nos.)	–	–	–	–	4	–	–	–	–	–	2	–	6	–	–	–	–	–	
Male (nos.)	–	4	–	–	1	–	–	1	–	–	–	–	–	–	–	–	–	–	
Female (nos.)	–	1	1	–	–	–	–	–	–	–	–	–	–	–	–	1	–	–	
(B) Sediment	
Mean (Xφ)	2.4	2.3	2.6	2.6	2.6	2.7	2.5	2.4	2.4	2.6	2.6	2.4	2.5	2.5	2.5	2.5	2.5	2.7	
Sorting (σφ)	0.8	0.8	0.6	0.5	0.6	0.5	0.7	0.8	0.7	0.5	0.7	0.6	0.6	0.7	0.5	0.5	0.6	0.5	
Skewness (SKφ)	−2.7	−2.2	−2.9	−2.5	−2.6	−2.5	−2.4	−2.7	−2.5	−2.4	−2.4	−2.2	−2.2	−2.4	−1.5	−2.0	−2.4	−2.4	
Kurtosis (Kφ)	11.3	8.7	16.2	15.9	14.6	16.2	10.9	12.0	10.9	15.0	11.5	11.3	12.6	11.4	9.4	12.3	13.5	16.8	
Gravel (%)	2.5	1.9	0.8	0.2	0.6	0.2	0.7	1.4	0.6	0.3	1.9	2.0	0.2	1.0	0.2	0.2	0.7	0.3	
Sand (%)	97.1	97.7	98.4	99.5	98.5	98.8	98.9	98.1	99.1	99.1	97.1	97.8	99.2	98.1	99.3	99.0	98.6	99.3	
Silt & Clay (%)	0.3	0.4	0.8	0.4	0.9	1.0	0.4	0.5	0.3	0.6	1.0	0.3	0.6	0.9	0.5	0.8	0.6	0.4	
0.125 mm (%)	37.1	28.7	50.4	43.7	41.3	47.5	39.3	37.8	36.2	39.8	39.8	25.9	37.6	33.7	32.2	35.9	35.6	41.0	
0.180 mm (%)	38.6	41.4	29.3	37.1	34.8	30.3	37.2	36.6	43.3	38.4	32.2	47.7	39.0	33.4	43.0	41.0	40.5	33.4	
0.250 mm (%)	8.9	13.7	5.1	6.7	7.5	3.8	8.0	8.8	6.5	7.3	7.1	15.3	9.1	10.0	14.5	11.2	10.7	5.8	
Moisture depth (cm)	5.8	6.1	6.0	6.3	4.3	4.5	6.6	9.2	9.9	3.7	4.4	3.2	6.6	6.6	6.8	9.0	8.7	8.8	
Temperature (°C)	29.7	30.2	30.0	26.4	28.5	29.2	34.0	33.2	32.5	34.0	33.3	32.7	31.5	31.9	31.3	38.5	36.3	37.0	
pH	6.5	6.6	6.7	5.8	6.0	5.8	2.5	2.2	2.8	6.2	6.4	6.3	4.8	4.8	4.3	3.5	4.2	5.0	
Total Organic Carbon (%)	0.1	0.2	0.2	0.3	0.3	0.3	0.1	0.1	0.1	0.2	0.1	0.1	0.1	0.1	0.1	0.0	0.1	0.1	
Cd (mg kg−1)	14.0	8.8	13.8	22.7	16.4	14.0	16.1	12.0	21.9	2.3	8.9	10.4	7.3	11.0	1.0	9.6	5.7	39.8	
Cr (mg kg−1)	20.3	12.5	30.3	23.9	14.3	18.9	14.4	15.5	9.2	28.8	25.1	14.8	13.1	13.3	16.2	38.1	16.4	26.4	
Cu (mg kg−1)	5.7	3.0	4.1	2.8	1.7	4.5	3.3	2.6	1.6	4.3	5.0	2.1	3.2	4.9	2.9	9.8	5.4	5.0	
Pb (mg kg−1)	8.9	6.0	6.7	6.8	6.4	9.1	5.7	4.8	4.7	27.7	26.8	21.4	24.8	15.1	37.8	21.0	23.1	37.1	
Se (mg kg−1)	19.7	25.8	31.7	24.6	25.3	16.2	17.3	25.4	28.0	15.5	17.0	18.9	29.0	15.2	25.1	30.2	26.9	40.2	
Zn (mg kg−1)	22.0	16.3	20.0	21.7	17.4	22.6	15.9	14.3	13.7	31.6	36.5	23.3	28.1	25.9	29.3	30.5	31.4	46.2	
(C) Water	
Temperature (°C)	29.8	29.7	29.8	29.5	29.7	29.5	29.1	28.9	31.2	30.3	29.6	30.0	29.4	29.6	29.5	30.5	31.4	31.6	
pH	6.0	6.2	6.4	8.3	8.3	8.4	7.0	7.5	8.1	8.1	8.1	7.6	8.1	8.1	7.8	8.7	8.6	8.0	
Salinity (‰)	36.0	35.9	36.8	33.2	33.0	33.1	27.4	28.4	33.5	4.1	4.4	5.4	12.4	12.6	13.9	22.0	23.2	26.2	
Dissolved Oxygen (mg l−1)	7.0	6.8	7.0	6.1	5.5	6.5	5.2	9.1	8.0	7.9	10.8	3.1	6.6	9.6	6.3	6.2	10.0	10.1	
NO2− (mg l−1)	0.0	0.0	0.0	0.0	0.0	0.0	0.0	0.0	0.0	0.0	0.0	0.0	0.0	0.0	0.0	0.2	0.2	0.1	
NO3− (mg l−1)	0.5	1.7	1.1	0.3	0.5	0.6	0.7	0.3	0.6	1.4	1.0	0.8	0.9	1.6	1.1	1.9	1.9	0.5	
PO43− (mg l−1)	0.3	0.2	0.2	0.4	0.4	0.3	0.1	0.1	0.1	0.2	0.2	0.1	0.3	0.5	0.3	0.3	0.3	0.2	
S2− (µg l−1)	2.0	3.0	1.3	41.0	33.3	9.0	5.3	7.3	7.7	48.0	50.3	54.3	25.3	40.3	11.0	13.0	27.3	18.7	
Chl-a (mg l−1)	0.2	0.3	0.2	0.5	0.6	0.4	0.5	0.4	0.6	0.6	0.5	0.5	0.4	0.3	0.2	0.5	0.6	0.5	
Notes.

“–” No sample observed at the time of investigation.

Table 6 Pair-wise statistical variations (F values based on One-Way ANOVA) within biological and environmental parameters in relation to their study phases, seasons, sampling sites and lunar periods.

	Biology	Sediment	Water	
	Egg/nest	Gravel	Silt & Clay	pH	TOC	Temperature	pH	DO	NO2−	NO3−	PO43−	Chl-a	
Full moon vs. new moon	1.96	0.47	0.16	0.05	0.06	0.002	0.24	1.59	–	0.25	0.04	0.002	
Phase-1 vs. Phase-2	4.51	0.40	–	–	#	20.12	3400*	280*	#	#	#	#	
Phase-1 vs. Phase-3	0.03	2.69	3721*	20.49*	#	2.90	33.98*	2.09	#	#	#	#	
Phase-2 vs. Phase-3	2.04	1.85	2.08	3.0	3.0	0.13	61.79	3.09	–	280.33*	1.23	3.76	
SW vs. NE	0.45	0.79	0.06	1.24	81.0*	2.15	0.80	0.02	1.0	25.19*	0.03	0.02	
SW vs. IM	0.44	6.22	0.32	0.49	0.16	3.96	1.65	0.25	0	0.06	0.46	0.36	
NE vs. IM	0.01	1.43	0.11	1.83	0.25	2.19	0.04	0.02	1.0	9.63	1.34	0.27	
Site-I vs. Site-II	1.03	0.35	0.25	0.01	5.0	0.16	0.10	0.03	–	0.05	0.02	0.03	
Site-I vs. Site-III	0.66	0.06	6.9 × 10−4	0.02	0.53	0.37	0.52	0.03	–	0.17	0.01	9.9 × 10−4	
Site-II vs. Site-III	2.22	0.10	0.29	0.01	0.004	0.05	0.14	7.9 × 10−4	–	0.20	0.03	0.05	
Notes.

Phase-1: 2009–2010, Phase-2: 2010–2011 and Phase-3: 2012–2013.

SW Southwest monsoon

NE Northeast monsoon

IM Inter-monsoon

TOC Total Organic Carbon

DO Dissolved Oxygen

NO2− Nitrite

NO3− Nitrate

PO43− Phosphate

S2− Hydrogen Sulphide

* P < 0.05

“–” Data not strong enough for statistical comparison.

“#” No Phase-1 observations.

Heavy metal concentrations at the three nesting sites have followed the order of Cr>Zn>Se>Pb>Cu>Cd for Phase-2 (Tables 3–5). In Phase-3, the increased concentrations of Se and Cd at Sites I-II (in the order of Se>Zn>Cr>Cd>Pb>Cu) and increased concentrations of Zn, Se, Cd at Site-III (in the order of Zn>Se>Cr>Pb>Cd>Cu) observed. However, in terms of metal induced enrichment at the sampling sites, only Cd and Se have shown their extremity (Tables 7–8). Also, the geo-accumulation index suggests a heavy to extreme contamination of both Cd and Se at all sampling sites, especially during 2011–2013 (Tables 7–8).

Table 7 Metal induced Enrichment Factor (EF) and Geo-accumulation Index (I geo ) at the three (SI-III) nesting sites of T achypleus gigas at Pantai Balok during Phase-2 (2010–2011) survey.

2010	
	June	July	August	October	
	I	II	III	I	II	III	I	II	III	I	II	III	
(A) EF	
Cd	85.6E	55.3E	112.9E	104.8E	90.1E	86.3E	144.9E	130.4E	123.3E	104.6E	138.6E	82.8E	
Cr	1.7D	1.3D	1.9D	2.0M	2.0M	1.7D	2.8M	2.5M	2.4M	3.2M	3.4M	3.2M	
Cu	0.5D	0.6D	0.3D	0.6D	0.9D	0.8D	2.6M	2.2M	2.3M	1.6D	1.6D	1.7D	
Pb	0.5D	0.4D	0.5D	0.7D	0.9D	1.0D	1.9D	1.7D	1.7D	2.1M	2.3M	2.1M	
Se	131.7E	100.5E	155.2E	136.1E	107.1E	144.0E	380.2E	293.9E	373.2E	327.1E	342.7E	348.0E	
Zn	0.9D	0.7D	1.0D	1.0D	0.8D	0.8D	1.8D	1.5D	1.6D	1.5D	1.6D	1.6D	
(B) Igeo	
Cd	3.3H	2.9MH	3.6H	3.3H	3.0MH	3.1H	3.1H	3.0MH	2.9MH	2.1MH	2.5MH	1.7MC	
Cr	−2.1U	−2.3U	−2.1U	−2.2U	−2.3U	−2.3U	−2.3U	−2.5U	−2.6U	−2.8U	−2.6U	−2.7U	
Cu	−3.4U	−3.0U	−4.2U	−3.4U	−2.9U	−2.9U	−2.0U	−2.2U	−2.1U	−3.3U	−3.2U	−3.1U	
Pb	−2.5U	−2.7U	−2.5U	−2.3U	−2.0U	−1.7U	−1.5U	−1.7U	−1.8U	−2.0U	−1.8U	−2.0U	
Se	3.5H	3.2H	3.5H	3.2H	2.7MH	3.4H	4.0H	3.6H	4.0H	3.2H	3.2H	3.3H	
Zn	−2.1U	−2.3U	−2.1U	−2.4U	−2.7U	−2.6U	−2.1U	−2.4U	−2.3U	−3.0U	−2.9U	−2.9U	
2011	
	March	April	May	June	
	I	II	III	I	II	III	I	II	III	I	II	III	
(C) EF	
Cd	616.0E	601.2E	481.4E	537.1E	579.8E	524.8E	439.0E	282.4E	147.9E	402.3E	254.6E	109.0E	
Cr	10.4S	9.4S	6.7S	6.9S	7.0S	5.9S	5.9S	5.1S	4.5M	4.4M	4.5M	4.4M	
Cu	1.1D	0.8D	0.4D	0.8D	0.9D	0.9D	1.1D	0.6D	0.3D	0.8D	0.5D	0.3D	
Pb	6.2S	4.2M	6.1S	4.2M	5.1S	5.2S	3.8M	3.6M	3.6M	2.8M	3.0M	3.1M	
Se	1121.2E	984.7E	839.5E	984.3E	1091.7E	1015.4E	353.6E	342.8E	345.9E	678.8E	691.4E	680.0E	
Zn	3.6M	2.5M	2.4M	2.5M	2.8M	2.6M	1.9D	1.7D	1.6D	1.7D	1.7D	1.7D	
(D) Igeo	
Cd	3.6H	3.9H	3.5H	3.9H	4.0H	3.7H	3.6H	3.0MH	2.2MH	3.8H	3.2H	2.0MC	
Cr	−2.0U	−2.2U	−2.4U	−2.2U	−2.2U	−2.5U	−2.4U	−2.5U	−2.7U	−2.5U	−2.4U	−2.4U	
Cu	−4.7U	−4.7U	−6.2U	−4.7U	−4.6U	−4.7U	−4.4U	−5.0U	−6.2U	−4.4U	−4.9U	−6.0U	
Pb	−1.4U	−1.5U	−1.2U	−1.5U	−1.3U	−1.3U	−1.7U	−1.7U	−1.6U	−1.8U	−1.6U	−1.5U	
Se	4.0H	4.2HE	3.8H	4.2HE	4.4HE	4.2HE	2.8MH	2.8MH	2.9MH	4.0H	4.1HE	4.1HE	
Zn	−2.7U	−2.8U	−3.1U	−2.8U	−2.7U	−2.9U	−3.2U	−3.3U	−3.3U	−3.0U	−2.9U	−2.9U	
Notes.

Cd Cadmium

Cr Chromium

Cu Copper

Pb Lead

Se Selenium

Zn Zinc

The superscript letters in the Enrichment Factor shows—D, deficiency to minimal enrichment; M, moderate enrichment; S, significant enrichment; E, extremely high enrichment. The superscript letters in Geo-accumulation Index shows—U, uncontaminated; MC, moderately contaminated; MH, moderate to heavy contamination; H, heavy contamination; HE, heavy to extreme contamination.

Table 8 Metal induced Enrichment Factor (EF) and Geo-accumulation Index (I geo) at the three (SI-3) nesting sites of T achypleus gigas at Pantai Balok during Phase-3 (2012–2013) survey.

	2012	2013	
	May	June	July	August	Feb	March	April	May	
	I	II	III	I	II	III	I	II	III	I	II	III	I	II	III	I	II	III	I	II	III	I	II	III	
(A) EF for full moon observations	
Cd	296.5E	199.4E	81.6E	523.6E	364.4E	339.5E	737.3E	408.9E	416.6E	213.3E	216.2E	292.4E	193.0E	269.6E	168.6E	215.0E	231.5E	149.7E	372.1E	156.0E	258.2E	–	–	–	
Cr	1.5D	0.7D	1.5D	3.0M	0.7D	1.1D	1.8D	3.1M	0.9D	0.7D	0.5D	0.8D	0.8D	0.6D	0.8D	0.9D	0.9D	1.3D	0.8D	1.3D	0.9D	–	–	–	
Cu	0.2D	0.1D	0.3D	0.2D	0.2D	0.2D	0.2D	0.4D	0.2D	0.5D	0.2D	0.3D	0.2D	0.3D	0.2D	0.4D	0.2D	0.4D	1.3D	0.4D	1.2D	–	–	–	
Pb	0.6D	0.6D	0.8D	0.8D	0.8D	0.7D	0.6D	0.6D	0.7D	0.9D	0.8D	1.0D	2.3M	1.1D	2.0M	1.0D	0.9D	1.3D	1.7D	1.1D	1.7D	–	–	–	
Se	396.9E	345.7E	390.0E	555.9E	418.0E	290.9E	547.2E	490.1E	343.3E	292.1E	280.8E	287.9E	216.9E	190.4E	419.8E	287.8E	239.9E	121.6E	253.5E	311.5E	279.1E	–	–	–	
Zn	5.3S	4.7S	6.6S	6.2S	6.4S	6.3S	6.4S	5.5S	5.9S	7.0S	6.9S	7.8S	7.0S	6.0S	7.4S	6.1S	6.2S	7.0S	6.6S	6.2S	8.0S	–	–	–	
(B) EF for new moon observations	
Cd	266.3E	210.4E	290.6E	544.4E	359.6E	317.7E	432.4E	309.2E	634.1E	–	–	–	30.9V	88.6E	218.7E	110.1E	174.6E	19.5S	–	–	–	114.9E	57.9E	572.1E	
Cr	1.1D	0.9D	1.8D	1.6D	0.9D	1.2D	1.1D	1.1D	0.8D	–	–	–	1.1D	0.7D	0.9D	0.5D	0.6D	0.9D	–	–	–	1.3D	0.5D	1.1D	
Cu	0.4D	0.3D	0.3D	0.3D	0.1D	0.4D	0.4D	0.3D	0.2D	–	–	–	0.2D	0.2D	0.2D	0.2D	0.3D	0.2D	–	–	–	0.5D	0.2D	0.3D	
Pb	0.8D	0.7D	0.7D	0.8D	0.7D	1.0D	0.8D	0.6D	0.7D	–	–	–	1.8D	1.3D	2.3M	1.8D	1.2D	3.5D	–	–	–	1.3D	1.1D	2.6M	
Se	263.2E	437.4E	457.6E	414.6E	396.7E	262.4E	332.7E	464.6E	574.9E	–	–	–	149.4E	125.4E	283.0E	300.2E	177.1E	351.8E	–	–	–	263.5E	182.6E	408.1E	
Zn	6.2S	6.0S	6.1S	7.7S	5.6S	7.6S	6.4S	5.5S	5.9S	–	–	–	6.2S	5.3S	7.4S	6.1S	6.2S	8.2S	–	–	–	5.6S	4.5S	9.7S	
(C) Igeo for full moon observations	
Cd	5.0HE	4.8HE	2.2MH	5.3EC	5.0HE	4.6HE	5.9EC	5.2EC	5.0EC	4.3HE	4.5HE	5.1EC	4.3HE	4.6HE	4.0HE	5.5EC	5.1EC	3.9H	5.3EC	4.4HE	4.8HE	–	–	–	
Cr	−2.2U	−3.0U	−2.9U	−1.8U	−3.7U	−3.3U	−2.6U	−1.6U	−3.7U	−3.6U	−3.8U	−3.1U	−3.3U	−3.5U	−3.3U	−2.1U	−2.7U	−2.3U	−3.1U	−2.2U	−3.1U	–	–	–	
Cu	−4.6U	−4.9U	−4.9U	−5.0U	−4.9U	−5.4U	−5.1U	−4.4U	−5.0U	−3.5U	−4.8U	−4.1U	−4.8U	−4.0U	−4.6U	−2.8U	−4.3U	−3.3U	−1.8U	−3.3U	−2.2U	–	–	–	
Pb	−2.2U	−1.9U	−2.6U	−2.4U	−2.0U	−2.6U	−2.9U	−2.7U	−2.6U	−1.9U	−1.9U	−1.5U	−0.4U	−1.2U	−0.6U	−0.6U	−1.2U	−0.9U	−0.5U	−1.2U	−0.8U	–	–	–	
Se	4.9HE	5.1EC	4.3HE	4.8HE	4.9HE	3.9H	4.9HE	4.9HE	4.2HE	4.3HE	3.5H	4.5HE	4.0HE	4.0HE	4.8HE	5.2EC	4.6HE	3.4H	4.5HE	4.9HE	4.4HE	–	–	–	
Zn	−2.9U	−2.8U	−3.3U	−3.3U	−2.9U	−3.3U	−3.5U	−3.3U	−3.3U	−2.8U	−2.6U	−2.4U	−2.6U	−2.6U	−2.6U	−1.8U	−2.3U	−2.3U	−2.4U	−2.4U	−2.4U	–	–	–	
(D) Igeo for new moon observations	
Cd	4.9HE	4.1HE	4.9HE	5.6EC	5.2EC	4.6HE	5.2EC	4.7HE	5.6EC	–	–	–	2.2MH	4.2HE	4.2HE	3.2H	4.4HE	1.1ME	–	–	–	4.3HE	3.5H	6.3EC	
Cr	−2.7U	−3.4U	−2.2U	−2.5U	−3.2U	−2.8U	−3.2U	−3.1U	−3.9U	–	–	–	−2.2U	−2.5U	−3.2U	−3.4U	−3.3U	−3.1U	–	–	–	−1.8U	−3.1U	−2.4U	
Cu	−3.6U	−4.5U	−4.0U	−4.6U	−5.3U	−4.1U	−4.4U	−4.7U	−5.4U	–	–	–	−4.0U	−3.8U	−5.1U	−4.4U	−4.2U	−4.6U	–	–	–	−2.8U	−3.7U	−3.8U	
Pb	−1.8U	−2.3U	−2.2U	−2.1U	−2.2U	−1.7U	−2.4U	−2.7U	−2.7U	–	–	–	−0.1U	−0.2U	−0.5U	−0.3U	−1.0U	0.3UM	–	–	–	−0.5U	−0.4U	0.3UM	
Se	4.3HE	4.4HE	5.0HE	4.7HE	4.8HE	4.1HE	4.0H	4.8HE	4.9HE	–	–	–	3.9H	3.9H	4.2HE	5.0HE	3.9H	4.8HE	–	–	–	5.0HE	4.9HE	5.4EC	
Zn	−2.7U	−3.1U	−2.8U	−2.7U	−3.0U	−2.7U	−3.2U	−3.3U	−3.4U	–	–	–	−2.2U	−2.0U	−2.6U	−2.3U	−2.5U	−2.3U	–	–	–	−2.2U	−2.2U	−1.6U	
Notes.

Cd Cadmium

Cr Chromium

Cu Copper

Pb Lead

Se Selenium

Zn Zinc

“–” no sample observed at the time of investigation.

The superscript letters in the Enrichment Factor shows—D, deficiency to minimal enrichment; M, moderate enrichment; S, significant enrichment; V, very high enrichment; E, extremely high enrichment. The superscript letters in Geo-accumulation Index shows—U, uncontaminated; UM, uncontaminated to moderately contaminated; ME, moderate contaminated; MH, moderate to heavy contamination; H, heavy contamination; HE, heavy to extreme contamination; EC, extremely contaminated.

Water characteristics

Although Phase-2 and Phase-3 observations indicated a general trend of higher salinity during SW monsoon (25.3–28.0‰), followed by IM (7.6–16.2‰) and NE periods (9.3–10.8‰) (Tables 3–5), Phase-1 reveals a euhaline (>30‰) condition for all three seasons (Tables 1–2). The seasonal differences in surface water temperature, pH and DO were non-significant (Table 6). Also, the nutrients—NO2−, NO3− and PO43−, and Chl-a (observed for Phases 2-3) showed non-significant differences in relation to the seasons, except by NO3− for SW vs. NE monsoon. The concentration of S2− was however high for IM (38.9 ± 25 µg l−1) (mean ± SD), followed by NE (29.4 ± 20 µg l−1) and SW (20.8 ± 25 µg l−1) in the order. Overall, the variations in water quality parameters with respect to full/new moon periods, sampling sites and seasons were insignificant (Table 6).

Figure 3 Principal Component Analysis (PCA) showing the % variance in sedimentological parameters in relation to –(A) Tachypleus gigas egg count, (B) season, (C) lunar period and, (D) sampling sites.

The numbers 1–3 indicate study phases in the present investigation (1, Phase-1: 2009-2010; 2, Phase-2: 2010-2011 and 3, Phase-3: 2012-2013). The circle in each panel represents correlation circle and the orientation of the environmental (sediment) lines approximate their correlation to the ordination axes. Abbreviated environmental parameters: Temp, Temperature; Sort, Sorting; Skew, Skewness; Kurt, Kurtosis; Grav, Gravel; S&C, Silt and clay; MD, Moisture depth; and TOC, Total organic carbon.

Correlation between biological and environmental parameters

The PCA drawn between biological and sediment parameters showed 49.7% variance along axis-1 (eigenvalue: 6.96), and 13.2% variance along axis-2 (eigenvalue: 1.85) (Fig. 3). The total egg count was primarily associated with Phases 1 and 3, whereas its fall in Phase-2 was evident (Fig. 3A). While Phase-1 egg yield was largely associated with medium sand, silt and clay and gravel, the Phase-3 egg yield had other parameters like mean grain size, fine sand, medium-fine sand, temperature, moisture depth, etc., and shows the impact of monsoon (Fig. 3B), lunar period (Fig. 3C) as well as the preferred sites for T. gigas nesting (Fig. 3D). The BEST analysis has indicated that mean grain size is strongly correlated with the egg laying capacity of T. gigas (ρ = 0.137). In the case of water quality, the PCA shows 46.8% variance along axis-1 (eigenvalue: 4.21), and 18.5% variance along axis-2 (eigenvalue: 1.67) (Fig. 4). In fact, majority of the water quality variables were associated with Phase-3 egg counts (Fig. 4A) that occurred largely during SW monsoon (Fig. 4B), full moon periods (Fig. 4C) and at Sites I and III (Fig. 4D). The BEST analysis indicated that both (water) temperature and NO3− were correlated with T. gigas nesting activity (ρ = 0.069). Finally, the impact of heavy metals was seen only on Phase-3 (Fig. 5A) in relation to SW monsoon (Fig. 5B), full/new moon phases (Fig. 5C) and largely at Sites I-II (Fig. 5D) (cumulative variance along axis-1 (eigenvalue: 2.01) and axis-2 (eigenvalue: 1.83): 64.1 %). The BEST analysis showed a strong correlation of Cr and Se with the observed egg yield (ρ = 0.032).

Figure 4 Principal Component Analysis (PCA) showing the % variance in water quality parameters in relation to –(A) Tachypleus gigas egg count, (B) season, (C) lunar period and, (D) sampling sites.

The numbers 1–3 indicate study phases in the present investigation (1, Phase-1: 2009–2010; 2, Phase-2: 2010–2011 and 3, Phase-3: 2012–2013). The circle in each panel represents correlation circle and the orientation of the environmental (water) lines approximate their correlation to the ordination axes. Abbreviated environmental parameters: Temp, Temperature; Sal, Salinity; DO, Dissolved oxygen; Chl-a, Chlorophyll-a; NO2−, Nitrite; NO3−, Nitrate; PO43−, Phosphate; S2−, Sulphide.

Discussion

State-of-the-art on Tachypleus gigas population at Pantai Balok

The impact of local environmental changes can easily be witnessed through declining or loss of biodiversity (Essl et al., 2015). Although Pantai Balok has been supporting the arrival of T. gigas for decades (BR Nelson, pers. comm., 2009 with local people), the increased anthropogenic disturbances in recent years (e.g., settlements, pollution) are severely affecting the spawning crab’s arrival (Tan et al., 2012; Azwarfarid et al., 2013). In fact, the present study also supports this scientific concern. After discovering that T. gigas stopped their spawning activity at Tanjung Selangor by 2011 (Nelson et al., 2015), we expected more crabs and higher spawning activity at Pantai Balok located ∼37 km away from Tanjung Selangor (in the north). But the present study indicates further decline in the T. gigas population due to sediment/water quality changes caused by physical infrastructure developments locally (during 2011–2013).

Figure 5 Principal Component Analysis (PCA) showing the % variance in heavy metals in relation to –(A) Tachypleus gigas egg count, (B) season, (C) lunar period and, (D) sampling sites.

The numbers 2–3 indicate study phases in the present investigation (2, Phase-2: 2010–2011 and 3, Phase-3: 2012–2013). The circle in each panel represents correlation circle and the orientation of the environmental (heavy metals) lines approximate their correlation to the ordination axes. Abbreviated environmental parameters: Cd, Cadmium; Cr, Chromium; Cu, Copper; Pb, Lead; Se, Selenium; Zn, Zinc.

Figure 6 Schematic diagram showing the nest shifting behaviour of Tachypleus gigas between the three sampling sites (I–III) at Pantai Balok (Google Maps©2013).

The impact of environmental processes (indicated in boxes) in relation to monsoon (SW/IM/NE) seasons are found to be important for spawning site(s) selection by the adult crabs.

The loss of horseshoe crabs can vary between genders. The male and female ratio of 31:38 in Phase-1 decreased to 13:7 in Phase-2 and 19:9 in Phase-3. More males can indicate ‘spawning strategies’ and ‘sperm competition’ (i.e., satellite behaviour and tandem mating) which is known to occur in the areas of small horseshoe crab assemblages (Schaller, Thayer & Hanson, 2002; Brockmann & Smith, 2009; Mattei et al., 2010; Brockmann & Johnson, 2011; Beekey & Mattei, 2015). On the other hand, overexploitation of the female crabs by local communities is also responsible (at least to some extent) for its reduced counts. The demand for female horseshoe crabs was reported previously from different countries like Thailand, India, America, Malaysia, etc. (Christianus & Saad, 2009; Basudev et al., 2013; Beekey & Mattei, 2015). Yet the negligible encrustation of fouling organisms on female T. gigas (with clear mating scars) at Pantai Balok shows the healthy nature of the organisms (cf. Brockmann, 2002). Also, their arrival at Balok beach (despite the anthropogenic disturbances) confirms this place as a choice and still supporting the environmental cues (Chabot & Watson, 2010), beach topography (Chatterji et al., 2008; Jackson & Nordstrom, 2009; Brockmann & Johnson, 2011), and sediment/ water characteristics (Chatterji & Shaharom, 2009; Smith et al., 2011; Zaleha et al., 2011; John et al., 2012; Nelson et al., 2015) that are necessary for T. gigas spawning. Therefore, appropriate conservation and management efforts would be able to retain not only this crab’s population, but also its final spawning ground on the east coast of P. Malaysia.

Factors influenced the spawning activity of Tachypleus gigas at Pantai Balok

At the beginning of this study (Phase-1), only sand mining was presumed to have an impact on water and sediment characteristics and affect the spawning activity of T. gigas. However, this specific concern couldn’t be compared with earlier studies/findings from Pantai Balok. But the later developments in the form of wave breaker/parking lot (Phase-2) and fishing jetty (Phase-3) construction provided a better understanding of the local environmental changes vis-à-vis T. gigas population dynamics. It was clear that the sediment close to spawning sites experienced a series of changes and finally became dominated by fine sand with moderate to well-sorting, very coarse skewed and extremely leptokurtic properties. However, the occurrence of more nests and eggs in Phase-3 was suggesting that fine sand, along with medium sand (of moderate sorting, symmetrical and very leptokurtic properties) as reported by Zaleha et al. (2012) and Nelson et al. (2015), also supports the spawning activity of T. gigas (mean grain size as strongly correlated). In addition, the average egg laying capacity of a female in Phase-3 (>400 nos.) was five times higher than to Phase-1 (80 nos.), and exceeded the counts (as max. 200–390 eggs) reported from Malaysia and India (Chatterji & Abidi, 1993; Chatterji, Kotnala & Mathew, 2004; Chatterji et al., 2004; Zaleha et al., 2012). In this context, the restricted distribution of the spawning crabs to the areas of high moisture depth and well oxygenated water was found to be crucial. The enhanced primary productivity in the estuaries due to nitrogen and its derivatives could attract not only lower tropic level organisms, but also other potentially important predators like horseshoe crabs which feed on a variety of prey (Carmichael et al., 2004). While the correlation observed between NO3− and T. gigas nesting is likely to benefit the adult crab forage (Carmichael et al., 2004; Haramis et al., 2007), the temperature could cause physical stress for the eggs (e.g., desiccation) (Ehlinger & Tankersley, 2004; Botton et al., 2006).

Although earlier researchers from Malaysia claimed year-round spawning activity of T. gigas (Zaleha et al., 2012; John et al., 2012; John, Jalal & Kamaruzzaman, 2013), the present study confirms it only between March and November, but covering all three seasons. However, there seemed to be a considerable influence of salinity on the egg laying capacity of female T. gigas. For instance, in Phases 2 and 3 when there was a general trend of high salinity during SW monsoon, the average number of eggs laid by a female was only 148–372 (despite the fact that SW favoured more spawning crabs, more nests and eggs). But with few observations of female crabs (n = 0 − 1), and if the total number of eggs is taken into account for other two seasons, there were as many as 556–967 for the NE monsoon and 32–360 for the IM period. While the maximum egg yield in the SW and IM periods is rather similar, it was remarkably high for the NE monsoon. This is however in contrast to the findings of Nelson et al. (2015) who reported no T. gigas spawning during the NE monsoon at Tanjung Selangor. Perhaps the differences in geographic location, beach topography, low/high impact of the seasonal water current (especially during NE monsoon) on spawning sites, etc., account for the differences encountered between the present study and Nelson et al. (2015). In the case of Phase-1, the euhaline conditions should have impacted the normal spawning activity of T. gigas (average no. of eggs laid by a female: 58 in SW, 50 in NE and 150 in IM). The poor sediment sorting which, in turn represents a weak water circulation/tidal mixing (Watanabe et al., 2014), very-fine skewed and extremely leptokurtic properties all support the observed euhaline situation. The narrowed river mouth in 2009–2010 was believed to be the main reason for this condition.

Horseshoe crabs avoid areas with vigorous tides and prefer relatively calm places for spawning (FitzGerald et al., 2008; Smith et al., 2011; Tan et al., 2012). At Pantai Balok, the spawning crabs preferred Site-I during the NE and IM periods, and Sites II and III during the SW monsoon. In South China Sea, the strong water current (with strong wind and waves) prevails during NE and IM (Camerlengo & Demmler, 1997; Akhir, Sinha & Hussain, 2011), and therefore the spawning crabs chose the interior location (i.e., Site-I) for their nesting. In fact, the spawning behaviour of the horseshoe crabs in relation to seasonal water current was explained by authors from India (Chatterji, Parulekar & Qasim, 1996), Malaysia (Nelson et al., 2015), America (Berkson & Shuster Jr, 1999), etc. Nevertheless, the horseshoe crabs are able to detect metal contamination and usually stay away from such locations (Itow, 1997a; Itow, 1997b). However, the extremity of Cd and Se (though found strongly correlated) in the sediment didn’t completely challenge the spawning crab’s arrival at Pantai Balok and so there is hope for its resurgence if the ongoing anthropogenic disturbances are regulated through appropriate conservation and management efforts.

Conservation and management measures for Tachypleus gigas at Pantai Balok

Due to poor understanding of the horseshoe crab population dynamics and breeding biology, several of its known and unknown spawning grounds are on the verge of extinction (Chiu & Morton, 2003; Ehlinger & Tankersley, 2007; Moore & Perrin, 2007; Cartwright-Taylor, Lee & Hsu, 2009). On the east coast of P. Malaysia, the current situation of halted spawning activity by T.gigas at Tanjung Selangor might also occur at Pantai Balok if no regulations on the beach construction are imposed. For instance, the concerned governmental and/or non-governmental authorities should conduct an “Environmental Impact Assessment” before starting any construction work at Pantai Balok. Also, appropriate care should be taken to prevent overexploitation of the crabs (especially females) for edible and fishmeal preparations. The awareness of the horseshoe crab’s significance and possible health hazards, if they are caught and consumed from polluted areas, must be communicated to the local people. Also, the standardized sampling protocols can develop a strong database for future scientific comparisons and effective long-term monitoring (cf. Botton, 2001; Sekiguchi & Shuster Jr, 2009; Shuster Jr & Sekiguchi, 2009). Although several law and enforcement rules are available for the protection of the forest reserves, biodiversity, etc., in Malaysia, the horseshoe crabs do not have gazetted status. In fact, the additional protection offered to the majority of animals under the Fisheries Act 1985 (Act 317) and the Wildlife Conservation Act 2010 (Act 716) in Malaysia did not include the horseshoe crabs so far. Therefore, it is mandatory to have a nation-wide priority on the conservation and management of these crabs. Perhaps community involved management efforts would deliver better results than merely imposing rules and regulations in the vicinity. For a quick start, the signboards indicating ‘prohibited horseshoe crab fishing’ at Pantai Balok are necessary. In addition, the knowledge sharing programs like ‘touch and feel’, ‘measure, tag and release’, etc., which were conducted in the places like Taiwan (Chen, Yeh & Lin, 2004; Hsieh & Chen, 2009) could be adopted and incorporated into the local conservation module. If the above propositions are followed then it is going to benefit not only T. gigas, but also the other two Asian horseshoe crabs (T. tridentatus and C. rotundicauda) existing in Malaysia.

Conclusions

The present study, with a wide array of biological and environmental observations, evaluated the major environmental factors that influencing T. gigas nesting at Pantai Balok. After discovering the severity of anthropogenic disturbances that finally stopped the spawning activity of T. gigas at Tanjung Selangor, the present paper intends to protect these crabs at Pantai Balok—the last spawning ground for them on the east coast of P. Malaysia. Both medium and fine sand compositions were found to be suitable for T. gigas spawning, provided that moderate to well-sorting, symmetrical to very-coarse skewed and very leptokurtic to extremely leptokurtic conditions exist. Although the SW monsoon received more crabs, the average egg yield from a female was found higher for NE followed by SW and IM, respectively. The differences in male/female ratio at the spawning sites were largely due to overexploitation of the female crabs for edible and fishmeal preparations. The spawning crabs have also shown a seasonal nest shifting behaviour by choosing Site-I (interior estuary) during NE and IM periods (to avoid strong water current), and Sites II and III for SW monsoon (mild water current). In light of the crabs’ choice to spawn at Pantai Balok (despite the increased anthropogenic disturbances and pollution in recent years), it is crucial to implement all possible conservation and management measures. Importantly, the horseshoe crabs should be placed under both Fisheries and Wildlife Conservation Acts to bring nation-wide attention as well as priority in Malaysia.

Supplemental Information

Data S1 Supplementary data

Click here for additional data file.

The present study was administratively supported by AKUATROP, INOS and the School of Marine Science and Environment (PPSMS) at the UMT. The authors wish to thank Mohammad Mokhtari and the two unnamed referees for their objective criticism and valuable suggestions.

Additional Information and Declarations

Competing Interests

Author Contributions

Field Study Permissions

Data Availability

The authors declare there are no competing interests.

Bryan Raveen Nelson conceived and designed the experiments, performed the experiments, analyzed the data, contributed reagents/materials/analysis tools, wrote the paper, prepared figures and/or tables, reviewed drafts of the paper.

Behara Satyanarayana conceived and designed the experiments, analyzed the data, contributed reagents/materials/analysis tools, wrote the paper, prepared figures and/or tables, reviewed drafts of the paper.

Julia Hwei Zhong Moh reviewed drafts of the paper, support during fieldworks.

Mhd Abdullah Ikhwanuddin reviewed drafts of the paper.

Anil Chatterji conceived and designed the experiments, reviewed drafts of the paper.

Faizah Shaharom contributed reagents/materials/analysis tools, reviewed drafts of the paper.

The following information was supplied relating to field study approvals (i.e., approving body and any reference numbers):

The present research work proposal on T. gigas was formally reviewed and approved by both the Institute of Tropical Aquaculture (AQUATROP) and the Post-Graduate Centre (PGC) of UMT. In addition, the local fishermen association was consulted to obtain their permission. In this context, the association has given a verbal permission and no written consent was deemed necessary.

The following information was supplied regarding data availability:

The raw data has been supplied as Data S1.

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
