# Peer review of "The final spawning ground of Tachypleus gigas (Müller, 1785) on the east Peninsular Malaysia is at risk: a call for action"

_PeerJ, doi:10.7717/peerj.2232_

## Round 0.1 · original submission · Major Revisions

Please consider all the suggestions in the revised version.

·

Basic reporting

The manuscript is well organized and presents valuable information about the spawning ground of Tachypleus gigas. However, some modification and further analysis are required before publication.

1. Information regarding the Ph.D. proposal (lines 133-137) seems unnecessary.

2. I suggest using abbreviations for environmental vectors in PCA graphs. the positions of environmental vectors are indistinct particularly in figure 3, where the most of the environmental vectors are clumped on the right side of the graph.

Experimental design

No Comments

Validity of the findings

From the PCA BiPlots (Figure 4) it is obvious that some environmental factors are correlated with each other (especially sediment parameters). Therefore, it is recommended to perform Pearson correlation between environmental parameters before concluding about significant environmental variables, as some environmental factors may affect indirectly.

Reviewer 2 ·

Basic reporting

This is a potentially very interesting paper describing the temporal variation of spawning activity in the Malaysian horseshoe crab T. gigas and making associations to environmental and anthropogenic factors. The language and structure of the ms is very good, but I have some principal concerns about the design of the analysis and the interpretation of the results.

First, the ms does not state any clear hypothesis (l. 106-112), which makes the presentation of the results and discussion unfocused and without any clear outcome.

One problem with the analysis is that there is so much variation in the overall data set (e.g. season, place, phases), so it is very difficult to analyse correlations between the variation in spawning and the environmental features. At the same time some of the important environmental pressures that may be responsible for the severe drop in abundance of individuals may not have been recorded (e.g. fishing activity, construction activity). The only obvious anthropogenic driver with relation to the measured variables is the sand mining. Hence, the statistical tests perform weak when trying to identify those conditions and factors that are optimal for TG.

The only clear pattern I see is that Phase 1 is very different from 2 and 3, both regarding number of females and regarding water quality. Sediment quality however, seems to be different among all three phases, so it may not be so clear to link this to the disappearance of females. Also, I would expect the sediment characteristics to affect the number of nests/eggs but not the number of females. However, the number of eggs has increased from phase 1/2 to phase 3. Again, what is missing here is a clear causal link between the potential pressure, the measured variable and the analysis of variance.

I also find it difficult to overview the ANOVA results from the text descriptions. Please consider moving all statistical test results into table(s), e.g. with columns for variables and rows for phase-pairs that are compared against each other.

It is confusing to use 1, 2, 3 for both sites and phases. I suggest changing the site description to site A, B, C.

Spell check for ‘carbs’ vs ‘crabs’

I still find the ms intriguing and a potentially very critical contribution to the field of horseshoe crab conservation. In conclusion I suggest major revision of the manuscript.

Experimental design

see above

Validity of the findings

see above

Additional comments

see above

Reviewer 3 ·

Basic reporting

Pass

Experimental design

Pass
HCl has been added to water samples to adjust the pH and same sample has been used to analysis of chlorophyll a. HCl affect for degradation of chlorophyll in samples. See the analysis of pheaopigments, Therefor chlorophyll data may not be accurate. Parsons, Maita&Lalli, 1984 have not mentioned the method which you have used for chlorophyll-a analysis.

Validity of the findings

Pass
Except chlorophyll data

Additional comments

Introduction
Cite references in text correctly. See the guideline
Line 59: (Rudkin and Young, 2009)
Line 71: Brockmann and Smith, 2009.
Line 74: (Chatterji and Abidi, 1993)
Line 76: Weber and Carter, 2009
Line 78: Chatterji and Shaharom, 2009
Line 83: Chabot and Watson, 2010)
Line 84 and 85: Chatterji andShaharom, 2009;
Line 77: Remove “eg”

Materials and Methods
Correct the Figure 1 caption. (F) has not mentioned and D is repeated
Fish jetty replace with fishing jetty
Line 128: Fish jetty replace with fishing jetty
Line 131: Chicken replace with chick
Line 133-139: This paragraph is not relevant to the materials and methods. It is about one of the authors registration for Ph. D and getting the permission for field work. Thus I suggest to remove this paragraph.

Sampling protocol
Correct caption of Figure 2 as below
Figure 2: Egg yield of Tachypleusgigasat PantaiBalok in relation to - (A) season, (B) lunar
period (C) sampling sites and, (D) The number of male and female spawning crabs arrived to
Balok beach.
Hydrological observations
HCl has been added to water samples to adjust the pH and same sample has been used to analysis of chlorophyll a. HCl affect for degradation of chlorophyll in samples. See the analysis of pheaopigments, Therefor chlorophyll data may not be accurate. Parsons, Maita&Lalli, 1984 have not mentioned the method which you have used for chlorophyll-a analysis.
Line 186: Replace “ 0.2 °C” with “±0.2 °C”
Line 187: 3” PVC tube. Is this diameter or radius?

Results
Spawning population and nesting
Authors have mentioned that the full moon observations revealed higher egg yield (7,261 eggs in 48 nests) than to the new moon observations (1,693 eggs in 16 nests). But P value shows that there is no significant deference in egg counts during full moon and new moon.Also Fig. 2B doesn’t show above variation.

Graphical representation of results is not very clear in 2A, B and C. I suggest to use egg count/nest for all A, B and C graphs.

Draw line graphs for 2C and D

Sediment characteristics
Results of One –way ANOVA are not clear.

Water characteristics
Line 272: Replace “P = 3.1E-6) with “p=3.1 *10-6)” Also check the p value

Line 273: Replace “NO2, NO3 and PO4” with “NO-2, NO-3and PO3-4”

Line 316, 217:Replace “Brockmann and Smith, 2009” with “Brockmann& Smith, 2009”
Correct :Brockmann and Johnson,2011; Beekey and Mattei, 2015).
Christianus and Saad, 2009; Beekey and Mattei,2015).
Line 324:Chabot and Watson, 2010)
Line 325:Jackson and Nordstrom, 2009; Brockmann and Johnson, 2011),
Line 326:Chatterji and Shaharom, 2009
Line 380:Camerlengo and Demmler, 1997
Line 384:(Berkson and Shuster, 1999),
394 (Chiuand Morton, 2003; Ehlinger and Tankersley, 2007; Moore and Perrin, 2007;
404 and 405:Sekiguchi and Shuster, 2009; Shuster andSekiguchi, 2009)
415 Hsieh and Chen, 2009)

Don’t include the results in discussion
Results are repeating in the discussion
Revise the discussion removing results and include them in the results section if necessary.
Eg: See below
Line 339-342: The importantobservation here is that the female crabs dug more number of nests (32) and released morenumber of eggs (3.977 nos. in 43 clutches) in Phase-3 than Phase-1 (23 nests and 3,025 eggs in28 clutches) or Phase-2 (9 nests, 1,952 eggs in 10 clutches).

Iine 359:monsoon (25.3-28.0‰), the average number of eggs laid by a female was only 148-372 nos.

Line 362 and 363: consideration for other two seasons, there were as high as 556-967 eggs for NE monsoon(salinity: 9.3-10.8‰), and 32-360 eggs for IM period (salinity: 7.6-16.2‰).

Line 347:(Chatterji and Abidi, 1993;
Line 352:Ehlinger and Tankersley, 2004;
Don’t cite tables and figures in discussion to explain results

See
Line 379 : (Fig. 6).

Line 398:correctgovernmental and non-governmental

---

## Round 0.2 · Minor Revisions

Your manuscript still needs some final modifications as noted by the reviewers.

Reviewer 2 ·

Basic reporting

I like the revised version. It is much better structured now. There are still some flaws that could be easily addressed before publications:

Introduction

The last paragraph of the introduction is still really bad. It has bad grammar and mixes present with past tense. It also lacks any clear description of the objectives. Your paper investigates the relations between nesting behavior and environmental as well as anthropogenic factors, and tries to identify major drivers that influence the nesting. That is not stated anywhere in the manuscript.

Discussion

What are possible functional explanations for that relationship between nesting and NO3? This should be discussed.

The authors suggest an MPA’s for the investigated area. But this may be unrealistic and also may take a long time. The knowledge from this analysis should enable the authors to give more specific advise to the authorities with regard to identifying the most harmful anthropogenic activities in the area and to which extend these should be regulated.

Minor issues
l. 102 & 409: avoid the term ’long-term observation’ – this is used when data are collected over >20 years
l. 365: clam => calm
l. 418: the statement on over exploitation is not supported by any data/analysis

Experimental design

see above

Validity of the findings

see above

Additional comments

see above

Reviewer 3 ·

Basic reporting

Pass

Experimental design

Pass

Validity of the findings

Pass

Except the Pair-wise statistical variations. Is it one-way ANOVA? I have doubt on this statistical analysis. Require clarification. Otherwise suggest to remove table 6.

---

## Round 0.3 · accepted · Accept

Thank you very much for improving your manuscript.